# Effects of the Duration of Ying Yang Bao Consumption on Hemoglobin Concentration in Infants and Young Children in Less Developed Areas of China

**DOI:** 10.3390/nu14214539

**Published:** 2022-10-28

**Authors:** Jing Feng, Yongjun Wang, Tingting Liu, Junsheng Huo, Qin Zhuo, Zhaolong Gong

**Affiliations:** 1Key Laboratory of Trace Element Nutrition of National Health Commission, National Institute for Nutrition and Health, Chinese Center for Disease Control and Prevention, Beijing 100050, China; 2Department of Clinical Nutrition, The First Affiliated Hospital of Shandong First Medical University, Shandong Provincial Qianfoshan Hospital, Jinan 250014, China

**Keywords:** Ying Yang Bao, hemoglobin level, anemia, infants and young children

## Abstract

Ying Yang Bao (YYB) is conventionally prescribed as a nutritional supplement to infants and young children (IYC) in less developed areas of China. However, whether 18-month YYB consumption is reasonable needs assessment. This study examined the influence of the duration of YYB consumption on hemoglobin (Hb) levels and anemia prevalence. Data from the Nutrition Improvement Project on Children in Poor Areas of China in 2018–2019 were used. Questionnaires were used to collect information on basic characteristics, dietary status, and YYB consumption. Propensity score matching (PSM) was used to balance confounders. Hb levels and anemia prevalence in IYC with different durations of YYB consumption were compared. After PSM, all covariates were well-balanced, and 1151 pairs of IYC were included in subsequent analyses. During the 1st–9th months of intervention, YYB effectively increased Hb levels and reduced anemia prevalence in the intervention group. During the 10th–18th months of intervention, Hb levels in the control group increased and anemia prevalence decreased, while Hb levels and anemia prevalence fluctuated in the intervention group. In conclusion, YYB was effective in improving nutritional status of infants, but had a limited effect in young children. Nutritional supplements with different quantities or nutrients should be considered for young children.

## 1. Introduction

Anemia is the most prevalent nutritional deficiency, affecting 280 million children (approximately 40% of children) worldwide [1]. Research shows that, in 2019, 18.8% of children under 5 developed anemia in China [2]. Among children under 5, the prevalence of anemia is highest in infants and young children (IYC) aged 6–23 months. The *Report on Nutrition and Chronic Diseases in China*, published in 2020, reported that the prevalence of anemia in 6- to 23-month-old IYC of China was 36.9% overall and up to 42.0% in rural areas [3]. Although anemia may result from various causes, iron-deficiency anemia (IDA) is the dominant type, according to available evidence; approximately 50% of individuals with anemia have iron deficiency [4]. Long-term anemia not only affects the growth and development of children but also causes irreversible damage to their cognitive and nervous systems [5,6]. In addition to the health burden, anemia has a considerable social and economic impact [7,8]. Anemia is partly due to the coexistence of multiple micronutrient deficiencies [9]. Reducing anemia prevalence is a universal effort to eliminate all forms of malnutrition.

High prevalence of maternal malnutrition, low rate of breastfeeding, unacceptable complementary feeding, and poverty are the four major causes of malnutrition among Chinese children. Among these, unacceptable complementary feeding is the prominent factor [10]. IYC aged 6–23 months need more nutrition and energy for growth and development than at any other time in their life [11], and anemia occurs when rapid growth in IYC exceeds the availability of dietary iron. Supplementation with various micronutrients is a low-cost measure for increasing hemoglobin (Hb) concentration. WHO recommends daily iron supplementation in IYC as a public health intervention to prevent anemia and iron deficiency [12]. Many low- and middle-income countries and territories have implemented nutrition improvement programs for children, such as multiple micronutrient supplementation (MMS), lipid-based nutrient supplements (LNSs), and micronutrient powders (MNPs), to prevent anemia [13,14,15].

Reducing anemia in IYC under 2 years of age in rural areas has always been the top priority in China’s nutrition improvement work. In 2001, Professor Chen Chunming conducted a 24-month intervention study on nutritious supplementation in five poor counties of China’s Gansu Province [10]. The study compared the effects of dietary supplementation and rice flour intervention on length, weight, Hb, and intelligence in infants aged 4–12 months, and confirmed the positive role of supplementation in improving physical and intellectual development of IYC. This was the earliest study on complementary food supplements in China. After years of research by Chinese nutritionists, a nutrition pack, named Ying Yang Bao (YYB) in Chinese, gradually formed. In 2008, the establishment of national standards for complementary food supplements stimulated implementation of about 15 nutrition intervention projects in poor rural counties in which YYB was applied to infants and young children as home fortification for complementary feeding [16,17]. In the same year, YYB was applied in the Wenchuan earthquake-stricken areas to reduce the anemia and acute malnutrition of affected IYC, and the intervention effect was accepted and supported by parents, grassroots public health personnel, and village doctors [18]. Since 2012, the Ministry of Health and the All-China Women’s Federation have jointly implemented a project, the Nutrition Improvement Project on Children in Poor Areas of China (NIPCPAC), to address child anemia. The project provides a free pack of YYB for IYC from 6 to 24 months of age daily in poverty-stricken counties in China with special subsidies from the central government [19]. Its coverage has been expanded year by year, from 100 counties in 2012 to all 832 poverty-stricken counties in 2019. The project covered more than 9.47 million infants aged 6–24 months in 21 provinces in western and middle regions by the end of 2019 [10].

YYB is a home-based complementary food fortification developed to promote nutrition among IYC in high-risk areas of China. Although YYB distributed in different areas do not have the identical nutritional content, all products conform to *National Food Safety Standard: Dietary Supplement (GB22570-2014)* [20]. The nutrition pack is composed of soybean powder or milk powder as the base, adding a variety of nutrients and other supplementary materials [20]. Each pack of YYB weighs 12 g and provides approximately 200 kJ energy on average. It contains 3.0 g protein, 200 mg calcium, 7.5 mg iron, 3.0 mg zinc, 250 μg RE vitamin A, 5.0 μg vitamin D, 75.0 μg folic acid, and some other vitamins and minerals [20]. Multiple studies in China have proved that the 18-month intervention with YYB can effectively reduce the underweight and wasting prevalence in IYC aged 6–24 months, and has a positive effect in reducing stunting and anemia related to malnutrition [17,21,22]. The nutrition community considers YYB a nutrition intervention suitable for IYC in developing countries [10]. The project team of the Nutrition Institute of China Center for Disease Control and Prevention conducted meta-analyses on the effects of YYB. The results showed that YYB consumption for 18 months, from 6 months to 24 months of age, could significantly increase Hb levels and reduce the incidence of malnutrition and anemia [23,24].

However, there remains concern about the effect of the duration of YYB consumption on Hb levels and anemia prevalence, and whether it is reasonable to consume YYB for 18 months. Will it be better if we supply YYB supplements to IYC for less or more than 18 months? To answer the above question, we explored the relationship between the duration of YYB consumption and Hb levels and anemia.

## 2. Materials and Methods

### 2.1. Study Design and Population

Study data were obtained from the Nutrition Improvement Project on Children in Poor Areas of China (NIPCPAC) in 2018–2019. As described in our previous article, this study used multistage sampling, probability proportional to size (PPS) sampling, and random equidistant sampling to select samples [11]. Through multistage random sampling, 9 of the 110 monitoring counties introduced to YYB were selected as intervention counties (Longhua, Moyu, Guiding, Tongyu, Yunlong, Songxian, Shilong, Yanchi, and Huangzhong), and 3 of the 40 not introduced in YYB were selected as control counties (Weichang, Luopu, and Fuquan) [25,26]. A pack of YYB was provided daily to IYC in intervention counties, whereas no pack was provided to those in control counties.

Individuals with infectious diseases, severe acute malnutrition, known developmental delay, or any other chronic disorders were excluded from recruitment. In addition, children with a known allergic reaction to soy protein or milk were excluded from intervention. Based on the sampling design and exclusion criteria above, approximately 410 IYC aged 6–24 months were selected from each county.

As shown in Figure 1, 4937 IYC aged 6–24 months were enrolled, and 4269 eligible IYC were assigned to either the YYB feeding group (YYB-FG, *n* = 3051) or the YYB non-feeding group (YYB-NFG, *n* = 1218) in this study.

### 2.2. Survey Indicators and Data Collection

Surveys were conducted by staff in county Maternity and Child Healthcare (MCH) hospitals. All staff had been trained by the Institute of Nutrition and Health of China Center for Disease Control. Caregivers of IYC aged 6–24 months old receive YYB from the local MCH hospitals regularly and are registered with them.

Questionnaires designed by NIPCPAC experts were used to collect data on basic characteristics and YYB consumption of IYC. Information of parents and caregivers, including age, education level, and occupation, was also obtained. The dietary status data were collected through a 24 h dietary recall of respondents [4,25]. The frequency of seven groups of complementary foods were collected: grains, roots, and tubers; vitamin A-rich fruits and vegetables; other fruits and vegetables; fresh foods (meat, fish, poultry, and liver/organ meats); eggs; dairy products (milk, infant formula, yogurt, and cheese); and legumes and nuts. Three indicators recommended by the World Health Organization (WHO) were used to evaluate the complementary feeding status of IYC: minimum dietary diversity (MDD), minimum meal frequency (MMF), and minimum acceptable diet (MAD) [27]. Moreover, the socio-economic status (SES) index was measured as a combination of parental education and occupation [28,29]. The educational level and occupational status of the parents reflects the socio-economic status of the family and the growth environment of IYC. SES is an indispensable confounder in this study.

Finger blood samples were obtained to measure Hb levels using HemoCue 301 (Angelholm, Sweden) during field investigation. Simultaneously, the Hb levels of infants before YYB consumption at 6 months were obtained from the MCH hospital online system. All infants had a physical examination in MCH hospitals at the age of 6 months, and the hospital’s electronic system will always retain information about their examination. Anemia was defined as Hb level of <110 g/L with adjustment for altitude, according to WHO criteria [30].

### 2.3. Data Handling and Analysis

EpiData (version 3.1; EpiData, Odense, Denmark) was used to establish a database and conduct double-entry verification. All data of individuals with missing key information were excluded from statistical analysis. IYC with low compliance (YYB consumption < 4 pack per week) in the intervention group were also excluded from the analysis [31]. Individuals with unknown YYB consumption or Hb level were also excluded.

To reduce the confounding effects, we used R software (version 4.1.2; “MatchIt” package) for propensity score matching (PSM). PSM is a statistical method that can effectively reduce some biases and confounding variables, thus allowing for a more reasonable comparison between groups [32,33]. Based on previous studies, the following 11 factors were considered as confounders: (1) baseline: age, sex, birth order, SES, education and occupation of caregivers [26]; (2) complementary feeding: MDD, MMF, MAD, and taking other supplements [11]; and (3) Hb levels of infants before taking YYB. To match as many pairs as possible, IYC from YYB-FG and YYB-NFG were 1:1 matched without replacement by the logit of the nearest propensity score, with a caliper of 0.10 times the standard deviation (SD).

Statistical analysis was performed using SPSS (version 19.0; IBM Institute, Albany, NY, USA). Data are reported as mean ± standard deviation for continuous variables and as number (frequencies) for categorical variables. Continuous variables were analyzed using paired t-tests, and categorical variables were compared by paired chi-square test. Differences were considered statistically significant at *p* < 0.05. R software (version 4.1.2; “cobalt” package) and Prism software package (version 9.0; GraphPad Software Inc., San Diego, CA, USA) were used for plotting.

### 2.4. Ethical Review

The project was reviewed and approved by the Ethics Committee of the Institute of Nutrition and Health of the Chinese Center for Disease Control and Prevention (No. 2018-017). All caregivers provided written informed consent.

## 3. Results

### 3.1. Basic Characteristics before and after PSM

We computed the standardized mean difference (SMD) between YYB-FG and YYB-NFG before and after PSM. Although there were slight differences in birth order between the two groups in the matched samples (*p* = 0.03), the SMD was set at <0.10 to indicate a good balance, and the Love plot confirmed no significant imbalance for any of the covariates included (Figure 2a). The mirrored histogram showed an adequate propensity score distribution and overlapping (Figure 2b).

Table 1 shows the distribution of baseline and complementary feeding factors in the original and matched samples. After PSM, there were 1151 matched pairs of IYC. Data for unmatched IYC in either group were not used in subsequent analyses. In matched samples, the Hb levels of infants before YYB consumption at 6 months were 118.72 g/L and 119.03 g/L in YYB-NFG and YYB-FG, respectively.

### 3.2. Effects of the Duration of YYB Consumption on Hb Levels and Anemia Prevalence

To ensure a balanced sample size for each group, we divided samples into six age groups. The Hb levels and anemia prevalence in IYC with different duration of YYB consumption are summarized in Table 2. During the 1st-9th months of YYB intervention, the trend of the Hb levels in YYB-NFG and YYB-FG was opposite: the levels decreased in YYB-NFG but steadily increased in YYB-FG, and the two groups reached their minimum and maximum of the entire intervention period at the 7th–9th months of YYB intervention, respectively. The Hb level was significantly higher in YYB-FG than in YYB-NFG at the 7th–9th months (Figure 3a). During the 10th–18th months of intervention, the Hb level in YYB-NFG increased, while fluctuating around 116 g/L in YYB-FG.

The prevalence of anemia in YYB-FG was lower than that in YYB-NFG at the early stage (1st–3rd months) and later stage (16th–18th months) of intervention, but there was no significant statistical difference. Its prevalence increased gradually in YYB-NFG during the 1st–9th months of YYB intervention and decreased in YYB-NFG. The prevalence of anemia in IYC was the highest (37.1%) in YYB-NFG while the lowest (19.6%) in YYB-FG at the 7th–9th months of intervention. The difference of anemia prevalence between the two groups was the largest at the 7th–9th months of intervention compared with that at other points of intervention (Figure 3b). During the 10th–18th months of intervention, the prevalence of anemia in YYB-NFG decreased gradually while fluctuating around 20% in YYB-FG.

## 4. Discussion

Anemia in IYC is still a major global public health problem [34]. According to *China’s National Program for Child Development (2021–2030)*, the prevalence of anemia in children under five should be controlled to less than 10% [35]. Infants and children aged 6–24 months in rural areas have always had a high incidence of anemia and have been the focus of nutrition workers [3]. YYB, a complementary food supplement, has helped to rapidly improve the nutrition status of infants and young children in poor rural regions in China [17]. Regarding the effect of YYB, Hb levels significantly improved and anemia prevalence reduced during the 1st–9th months of intervention, which is consistent with previous findings [23,36,37]. However, the effect in the 10th–18th months of intervention was not as effective as that in the 1st–9th months. There are also studies that disagree with our findings that the intervention was not effective in the 10th–18th months of intervention. Huo et al. claimed that the duration of YYB consumption correlated positively with Hb levels and negatively with anemia prevalence among IYC aged 6–23 months [26]. However, their study only demonstrated an impact of 6-month YYB consumption, without a parallel placebo or no-treatment control; therefore, conclusions cannot be drawn about a long-term influence. A systematic review of 27 research studies reported that LNSs, if provided for over 12 months, were more effective at improving growth and reducing anemia prevalence [38]. LNSs may contain sufficient nutrients to be effective in older children, and YYB differs from the LNS in the region and population.

On one hand, the intervention in the latter 9 months was not as effective as that in the first 9 months, possibly because the nutrients of YYB could no longer meet the rapid growth needs of IYC. The largest differences in Hb levels and anemia prevalence between the intervention and control groups were found at the 7th–9th months of intervention, when the children were 13–15 months old. Children aged 13–23 months require more energy and nutrients for healthy growth and development than infants aged 6–12 months [39]. In the study of Christian et al., they provided a small quantity of lipid-based nutritional supplements to children aged 6–12 months (125 kcal/day) and a medium quantity of supplements to children aged 13–18 months (250 kcal/day), guided by the energy intake recommended for preventing undernutrition [40]. The intervention did not differ by age in our study. For example, 10 mg Fe/day is recommended for infants aged 6–12 months, and 12 mg Fe/day is recommended for children aged 13–23 months [41]. According to the *National Food Safety Standard of Complementary Food Supplements (GB 22570-2014)*, 3.0~9.0 mg Fe/packet is recommended in supplements for infants aged 6–12 months, and 4.6~10.8 mg Fe/packet is recommended for children aged 13–23 months. An amount of 7.5 mg of iron per packet is close to the upper limit of what is recommended in supplements for infants aged 6–12 months, but close to the lower limit of what is recommended for children aged 13–23 months [20].

On the other hand, when IYC’s Hb levels are already within normal physiological levels, YYB may not be able to increase Hb levels further. At the 10th–18th months of intervention, there was little difference in the anemia prevalence between the intervention and control groups. The prevalence of anemia in this study population was significantly lower than the reported national average in rural areas [3]. YYB is mainly used to help IYC recover from nutritional anemia, avoid undernutrition, and maintain health [19,37]. Iron, zinc, and folic acid in YYB are all essential nutrients required for the maintenance of basic physiological function of red blood cells [42,43,44]. However, when these nutrients are sufficient, the body will not functionally respond to low-dose supplementation with additional elements [45]. Preventive zinc supplementation just has a small positive impact on growth [9].

The control group showed some improvement in Hb levels during the last 9 months of monitoring. This may have a lot to do with children getting older. Research has shown that young age is a factor associated with a high prevalence of anemia [4]. In the study of Liu et al., the prevalence of anemia in 6- to 11-month-old IYC was found to be the highest among all the age groups, and the prevalence of anemia decreased with age [4]. Moreover, it is possible that caregivers of children in YYB-NFG sought medical care on their own when they became aware of the child’s health problems, which influenced our study. In addition, studies have suggested that catch-up growth caused by early poor development may also affect later hemoglobin levels [46].

However, this study was limited by its cross-sectional design. This design demonstrates the association between YYB consumption and Hb and anemia, rather than the causality. There are other limitations to our study. First, PSM greatly reduced the sample size. The 1:1 matching used in this study caused the loss of many intervention samples. If 1:2 matching was selected, part of the controls would be lost due to unmatching. Second, unmeasured factors, such as access to medical care and better diet, may have influenced the results. In this study, complementary feeding and other supplements were included as confounders and adjusted by propensity score matching. However, we only considered whether the type and the frequency of food were acceptable. We did not take the amount of food consumption into account as we did not collect data on this, nor can we assess whether a diet is balanced. Third, this study only observed the effect of nutritional interventions on improving hemoglobin levels and reducing anemia, without taking other causes of anemia into account. Diagnosis of IDA requires the combination of such indicators as serum ferritin and soluble transferrin receptor, which were not detected in this study. We only judged anemia by the hemoglobin levels.

Our study also has its strengths. We were the first to look at the effect of different durations of YYB consumption on improving Hb levels and questioned whether it was reasonable to consume YYB continuously for 18 months. More importantly, our findings suggested that YYB may be less effective at improving nutrition status in young children, which has never been reported before. Additionally, this study not only considered the possible impact of family conditions but also considered the complementary feeding status on the nutritional status of IYC. Propensity score matching was used to effectively control known confounding factors [32].

## 5. Conclusions

YYB was effective in improving Hb levels and reducing anemia prevalence in IYC aged 6–14 months, but had limited effect in children aged 15–23 months. Perhaps, a 9-month intervention with YYB for IYC aged 6–14 months would be more reasonable. Nutritional supplements with different quantities or nutrients may be considered for IYC of different ages. YYB may be only beneficial in infants, and policies regarding supplementation in young children may need to include different amounts of nutrients. In addition, few cohort designs have been conducted on the effects of YYB, and more rigorous studies are needed.

## Figures and Tables

**Figure 1 nutrients-14-04539-f001:**
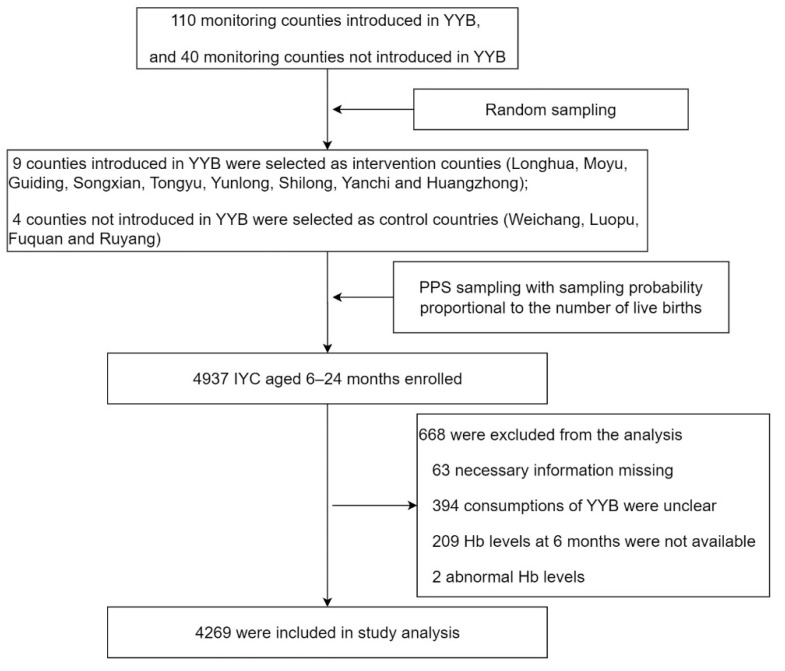
Flow chart. Inclusion and exclusion processes for subjects.

**Figure 2 nutrients-14-04539-f002:**
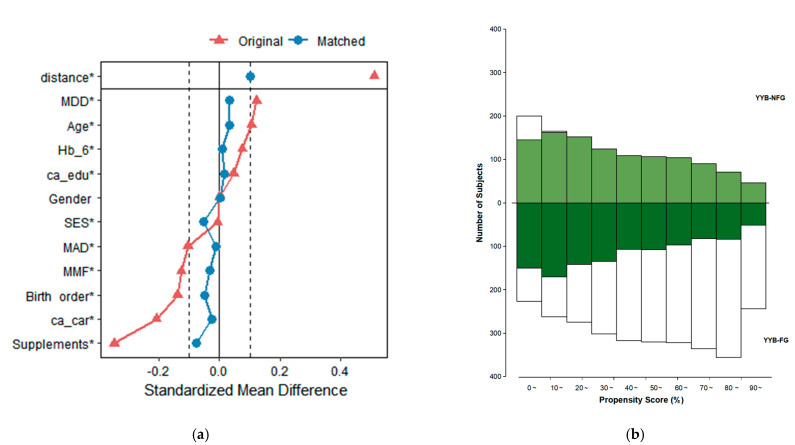
Assessment of covariate balance. (**a**) The Love plot shows changes in standardized mean difference before (red) and after (blue) propensity score matching. (**b**) The Mirrored Histogram shows the propensity score distribution and overlapping in unmatched (white) and matched (green) samples in the control (top) and treatment groups (bottom). SES: socio-economic status; ca_edu: education of caregivers; ca_car: career of caregivers; MDD: minimum dietary diversity; MMF: minimum meal frequency; MAD: minimum acceptable diet; Hb_6: hemoglobin levels of infants before YYB consumption at 6 months; YYB-FG: Ying Yang Bao feeding group; YYB-NFG: Ying Yang Bao non-feeding group. * Differences before and after propensity score matching were statistically significant at *p* < 0.1.

**Figure 3 nutrients-14-04539-f003:**
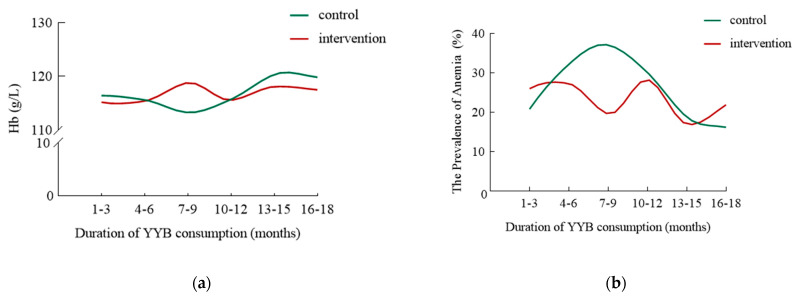
Hb level and the prevalence of anemia of IYC with different duration of YYB consumption. (**a**) The line chart shows the Hb of IYC in the control group (green) and the intervention group (red) at different stages. (**b**) The line chart shows the prevalence of anemia in the control group (green) and intervention group (red) at different stages. Hb: hemoglobin levels; YYB: Ying Yang Bao; YYB-FG: Ying Yang Bao feeding group; YYB-NFG: Ying Yang Bao non-feeding group.

**Table 1 nutrients-14-04539-t001:** Characteristics of infants and young children in the original and matched samples.

	Original			Matched		
	YYB-NFG (n = 1218)	YYB-FG (n = 3051)	*p*	YYB-NFG (n = 1151)	YYB-FG (n = 1151)	*p*
Age (months), mean ± SD	16.39 ± 5.52	16.91 ± 4.97	<0.01	16.52 ± 5.48	16.51 ± 5.14	0.94
SES, mean ± SD	49.09 ± 7.29	49.05 ± 7.16	0.86	49.08 ± 7.28	48.83 ± 7.42	0.41
Birth weight (g), mean ± SD	3270.91 ± 505.83	3265.37 ± 475.56	0.74	3278.20 ± 504.71	3264.91 ± 471.37	0.51
Birth height (cm), mean ± SD	50.09 ± 1.64	50.10 ± 1.49	0.79	50.11 ± 1.63	50.06 ± 1.48	0.38
Hb_6 (g/L), mean ± SD	118.64 ± 9.00	119.44 ± 10.49	0.01	118.72 ± 8.95	119.03 ± 10.40	0.44
Gender, *n* (%)			0.97			0.68
Male	626(51.4)	1566(51.3)		597(51.9)	607(52.7)	
Female	592(48.6)	1485(48.7)		554(48.1)	544(47.3)	
Birth order, *n* (%)			<0.01			0.03
First-born	406(33.4)	1230(40.4)		394(34.3)	398(34.6)	
Non-first-born	809(66.6)	1817(59.6)		754(65.7)	752(65.4)	
Parents as main caregivers, *n* (%)	992(81.4)	2438(79.9)	0.25	934(81.1)	945(82.1)	0.55
Caregiver’s education, * n * (%)			0.01			0.29
Primary School or Below	288(23.8)	751(24.7)		271(23.7)	289(25.1)	
Middle School	709(58.5)	1644(54.0)		667(58.3)	634(55.1)	
High School or Above	214(17.7)	647(21.3)		206(18.0)	227(19.7)	
Caregiver domestic unemployed, *n* (%)	715(59.1)	2093(69.0)	<0.01	699(61.1)	711(61.9)	0.70
Consume other supplements, *n* (%)	372(30.6)	534(17.7)	<0.01	311(27.1)	307(26.8)	0.86
MDD, *n* (%)	899(75.2)	2445(81.1)	<0.01	863(76.2)	864(76.2)	0.99
MMF, *n* (%)	468(38.8)	1042(34.2)	<0.01	433(38.0)	448(39.0)	0.62
MAD, *n* (%)	343(29.0)	798(26.5)	0.11	321(28.6)	342(30.2)	0.40

SES: socio-economic status; Hb_6: hemoglobin levels of infants before taking YYB at 6 months; MDD: minimum dietary diversity; MMF: minimum meal frequency; MAD: minimum acceptable diet; YYB-FG: Ying Yang Bao feeding group; YYB-NFG: Ying Yang Bao non-feeding group.

**Table 2 nutrients-14-04539-t002:** Hb levels and anemia prevalence of IYC with different duration of YYB consumption.

Duration of YYB Consumption (Months)	Hb (g/L), Mean ± SD			Anemia, *n* (%)		
	YYB-NFG	YYB-FG	*p*	YYB-NFG	YYB-FG	*p*
1–3	116.44 ± 9.69	115.19 ± 11.84	0.33	32(20.8)	40(26.0)	0.36
4–6	115.64 ± 11.19	115.42 ± 11.23	0.86	46(32.2)	39(27.3)	0.43
7–9	113.29 ± 13.67	118.83 ± 11.34	<0.01	53(37.1)	28(19.6)	<0.01
10–12	115.75 ± 12.95	115.58 ± 12.12	0.89	52(30.1)	49(28.3)	0.82
13–15	120.37 ± 12.38	118.10 ± 10.98	0.03	42(18.8)	38(17.0)	0.70
16–18	119.84 ± 12.49	117.51 ± 11.48	0.01	51(16.2)	69(21.9)	0.08

## Data Availability

Not applicable.

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
