# Peer review of "Effects of the Duration of Ying Yang Bao Consumption on Hemoglobin Concentration in Infants and Young Children in Less Developed Areas of China"

_nutrients, 2022, doi:10.3390/nu14214539_

Round 1

Reviewer 1 Report

Well written paper. Key aspects are missing. I recommend adding suitable justifications prior to acceptance. Few comments and corrections. 

1) Replace anemia by iron deficiency anemia throughout the paper. Since these are rural areas, critical to identify the cause of anemia. 

2) Lines 40, 62, 183 - Need proper citation

3) Line 49: missing period

4) Lines 79, 81: Replace 'countries' with 'counties'

5) Line 99: Explain the hemocue blood collection process. Were the first few drops ignored?

6) Line 109: Remove spacing after bracket and move reference to the end of the sentence

7) What is SES and PSM? Introduce abbreviations.

8) Table 2: Justification is needed why Hb rose between 9 mo to 12 mo in the NFG group and decreased in the FG group. Similar vein, how did NFG ended up having higher Hb than FG? Seems like a negative outcome

9) Were tests done to ensure anemia was due to iron deficiency, and not chronic disorders? Any inflammatory markers measured such as CRP, serum ferritin, etc

10) Did you track what the NFG group was fed? Dietary assessment?

11) Lines 179-181: Half truth. What about 9-18 months?

12) Lines 190-192: What aged kids are recommended YYB in China?

13) Lines 194-195 : Parul Christian? Typo?

14) Line 200: 7.5 mg Fe/packet for infants. Isn't RDA for 6-12 mo around 10-11 mg/d?

15) Lines 207-210: Iron dietary excess can sometimes lead to iron overload. Not regulated in every situation.

16) Lines 219: What about access to better and balanced diet, more meat consumption, or inclusion of iron promoters? 

17) Line 224: "this has never been studied before", and yet, in line 182 it is mentioned that YYb was not effective? How is yours the first study then?

Author Response

Dear reviewer,

Thank you for your letter and comments concerning our manuscript entitled "Effects of the duration of Ying Yang Bao consumption on hemoglobin concentration in infants and young children in less developed areas of China"(nutrients-1968415). Based on your suggestions, we have accordingly revised our manuscript. Attached you could find the point-to-point responses to the questions regarding the manuscript.

We hope that our answers have satisfied your comments and look forward to your response.

Warm regards,

Reviewer 2 Report

This manuscript reports the results of an assessment of a nutritional supplement for iron deficiency in children, Ying Yang Bao, which is apparently common in rural China, but not well known to western clinicians and nutritionists.  Thus, it is important to share the information is important about how iron deficiency anemia is typically addressed in rural China.

The authors are forthcoming about the results, which apparently indicated there was some effectiveness for the first 9 months of the trial, which then waned during the last 9 months of the assessment.

There was some improvement in the control group during the latter months, which further complicated the conclusions about the long-term benefits of Ying Yang Bao.

The authors were also forthcoming about this methodological limitation in the Discussion section.

Because I personally was not previously aware of this particular supplement, I would ask the authors to include a statement about how they would compare the effectiveness of Ying Yang Bao to other commonly used iron supplements in other countries, and perhaps in urban China.  It certainly seems possible to remedy iron deficiency more rapidly with the many different sources of iron that be be administered now, assuming that was the goal.  I appreciate that there is also the question of parental acceptance of new interventions if there is a strong cultural preference and tradition of using Ying Yao Bao.

If one were being critical, one could ask if there was a constant source of the Ying Yang Bao across the trial?  For example, did they confirm that the iron concentrations remained the same across this entire time period?  They also state in the introduction that this supplement may be soy-based or milk-based.  The bioavailability of the iron in the supplement for absorption by the gut would likely differ between soy and milk, especially if the milk powder was derived from cow's milk.

The authors might also include one additional comment about the causes of the high levels of iron deficiency in Chinese infants beyond under-nutrition, or a low level of iron supplementation in maternal diets during pregnancy.  Some environmental pollutants, especially lead and zinc, can interfere with iron absorption in infants.  Lead pollution is a concern in the urban, industrial areas, and high levels of lead have been reported in many Chinese infants.  They don't mention if the children in this study had been tested for trace metals.

Notwithstanding these questions, the information about this Chinese supplement is of interest.  One recognizes that it may not be possible to modify a reliance on a traditional supplement even if other more effective ones were available.  In addition, there is the fiscal constraint of cost.  Perhaps the authors would include a comment on whether Ying Yang Bao is economical and more easily afforded by these families that other supplements.

Author Response

(The authors gave the same response as above.)

Round 2

Reviewer 1 Report

Edits have been addressed.